# Establishment of Monterrey Pine (*Pinus radiata*) Plantations and Their Effects on Seasonal Sediment Yield in Central Chile

Roberto Pizarro [1,2,3,4], Pablo García-Chevesich [5,6,*], Ben Ingram [7], Claudia Sangüesa [1,2], Juan Pino [8], Alfredo Ibáñez [1,2], Romina Mendoza [2,3], Carlos Vallejos [2], Felipe Pérez [9], Juan Pablo Flores [10], Mauricio Vera [2], Francisco Balocchi [11,12], Ramón Bustamante-Ortega [2] and Gisella Martínez [13]

1 Centro Nacional de Excelencia para la Industria de la Madera (CENAMAD), Pontificia Universidad Católica de Chile, Santiago 7810128, Chile
2 UNESCO Chair Surface Hydrology, University of Talca, Talca 3467769, Chile
3 Instituto Interdisciplinario para la Innovación, Universidad de Talca, Talca 3467769, Chile
4 Faculty of Forest Engineering and Nature Conservancy, University of Chile, Santiago 8820808, Chile
5 Department of Civil and Environmental Engineering, Colorado School of Mines, Golden, CO 80401, USA
6 Intergubernmental Hydrological Programme, UNESCO, Montevideo 11200, Uruguay
7 School of Water, Energy and Environment, Cranfield University, Cranfield MK43 0AL, UK
8 Dirección de Transferencia Tecnológica, Universidad Tecnológica Metropolitana, Santiago 8330367, Chile
9 Dirección General de Aguas, Ministerio de Obras Públicas, Santiago 8340652, Chile
10 Centro de Información de Recursos Naturales (CIREN), Santiago 7501556, Chile
11 Ecosystems, Productivity and Climate Change, Bioforest SA, Camino a Coronel km 15, Coronel 413000, Chile
12 Water Resources and Energy for Agriculture PhD Program, Water Resources Department, Universidad de Concepción, Chillán 3812120, Chile
13 Facultad de Geología y Minas, Universidad Nacional de San Agustín de Arequipa, Arequipa 04000, Peru
* Correspondence: pablogarcia@mines.edu

**Abstract:** Sediment production and transport in a basin are generally a function of the degree of soil protection, normally represented by plant cover. In this study, two basins located at similar latitudes but with different hydrological regimens and plant covers were studied, one with a pluvial regimen and forest plantations (Purapel) and another one with the pluvio-nival regimen and native forest (Ñuble). For this purpose, sediment yield was analyzed in both drainage areas using the Mann-Kendall statistical test. Both basins showed larger amounts of sediment production during winter months. In addition, sediment yield trends did not show significant variation in the case of the Ñuble, most likely due to non-relevant changes in plant cover over time. However, there is a sustained decrease in annual sediment release at Purapel, coinciding with the afforestation in the basin, so it is logical to attribute the referred reduction to this process. For the first time, the behavior of two watersheds is contrasted, one covered with native forest and the other one with forest plantations, appreciating that the basin covered with plantations presents a reduction in sediment production over time, which means that forest plantations are efficient in sediment retention, even in contrast to native forest. However, both basins have different types of soil, topography, etc., meaning that more studies are needed to support this theory.

**Keywords:** sediment yield; forest plantations; Chile; Monterrey pine; native forest; seasonal analysis





## 1. Introduction

Suspended sediments are a natural component of rivers and are related to the stability of contributing areas' ecosystems. As expected, sediment load transported during a few extreme events depends on soil erosion, plant cover, and rainfall characteristics, among others [1–3]. Despite being a natural process [4], sediment concentration in river flows is an important variable for the design of hydraulic structures, flood mitigation, river management, ecosystem conservation, and environmental protection [5]. This process may be accentuated in basins with higher slopes or intense storms [6,7]. However, anthropic

action can increase sediment yield by intervening basins due to the construction of roads, bridges, and culverts [8] and even population growth, e.g., [9], among others.

Sediment–discharge relationships vary across space and time and are vulnerable to human activities (e.g., land use changes and soil and water conservation techniques) and weather events [10]. Studies by Zhang et al. [9] concluded that the dynamics of rainfall–sediment load have changed in the context of ecological restoration. The study of the relationship between discharge (Q) and sediment concentration (C) has been used to evaluate different soil and water conservation measures in, for example, the Loess Plateau in China [10]. However, the effect of vegetation management is generally shown to be a lag of years for vegetation recovery and accumulation of plant cover area [9].

Additionally, suspended sediments fill reservoirs and affect water quality [11], increasing maintenance costs for these structures. Moreover, different measurement methods of quantification have been developed to estimate suspended load in rivers. In Chile, for example, the General Water Directorate (DGA) has stations in some rivers within the country to monitor monthly sediment loads. This information is used to analyze sediment concentration trends and generate mathematical relationships between C and circulating streamflow [12,13]. Among other aspects, this is not always possible because of the nature of the data, as some data are collected every month and others daily; for example, occasionally, sediment load is measured on a daily or sub-hourly basis [14].

Another factor to consider in sediment yield is the interaction between the soil–vegetation complex and basin hydrology, generally higher sediment concentration in basins with degraded terrains [15,16]. In this respect, plant cover decreases the amount of precipitation that impacts the soil directly by interception, thus reducing erosion rates [17–19]. Similarly, living or dead cover (litter) decreases rainfall's kinetic energy, increasing the resistance to the flow's surface flow, favoring infiltration, and minimizing sediment yield [15,20].

In the above context, the type of plant covers existing or predominant in a basin is another significant aspect to consider; regarding precipitation–runoff processes and their relation to C, different plant cover types should generate different C values. Thus, there is a widespread belief in public opinion that native forests would be more efficient in retaining sediments than forest plantations, e.g., [21]. However, other authors argued that there would be no significant differences, e.g., [22,23]. Moreover, other variables arise that could be affecting sediment processes, such as geology and seasonal detachment and transport of sediments. Central Chile has a Mediterranean climate with precipitation concentrated in winter months [24]; therefore, there should be an increase in sediment yield in winter, compared to summer, when rainfall shows significant decreases in quantity and intensity. However, snowmelt could also be affecting sediment production, especially during the summer months, which is an important process to analyze and study in pluvio-nival basins.

In this context, the precipitation–plant type–runoff process and its effects on C must be evaluated. According to C, information obtained from monitored basins, the objective of this research, this element could provide more knowledge concerning the efficiency of different types of plant covers. Therefore, the hypothesis that motivates this research is that sediment yield is the result of a multifactorial process, i.e., the production of sediment depends in part on the following factors: soil erodibility, rainfall intensity, slope, agricultural activities, and plant cover, among others, in which plant cover could be a determining factor.

## 2. Materials and Methods

The study was developed in two basins, both located in central Chile (Figure 1). The first basin (665 km$^2$) is the Purapel River in Sauzal ("Purapel"), located in the La Costa mountain range (Maule Region, low elevation coastal area). This basin has a regimen completely pluvial (i.e., rainfall-fed) and a Mediterranean climate (i.e., there is a prolonged dry season during summer months, which in some years has caused the Purapel River to dry out completely). Unlike the Purapel basin, the second unit of study corresponds to the Ñuble in San Fabián river basin ("Ñuble"), a 1654 km$^2$ basin located in the Andean foothills of the Ñuble Region and is located under the Mediterranean climate. This basin has a

pluvio-nival regimen (i.e., rainfall- and snowmelt-fed), usually presenting two periods of high streamflow (winter due to rainfall and summer due to snowmelt).

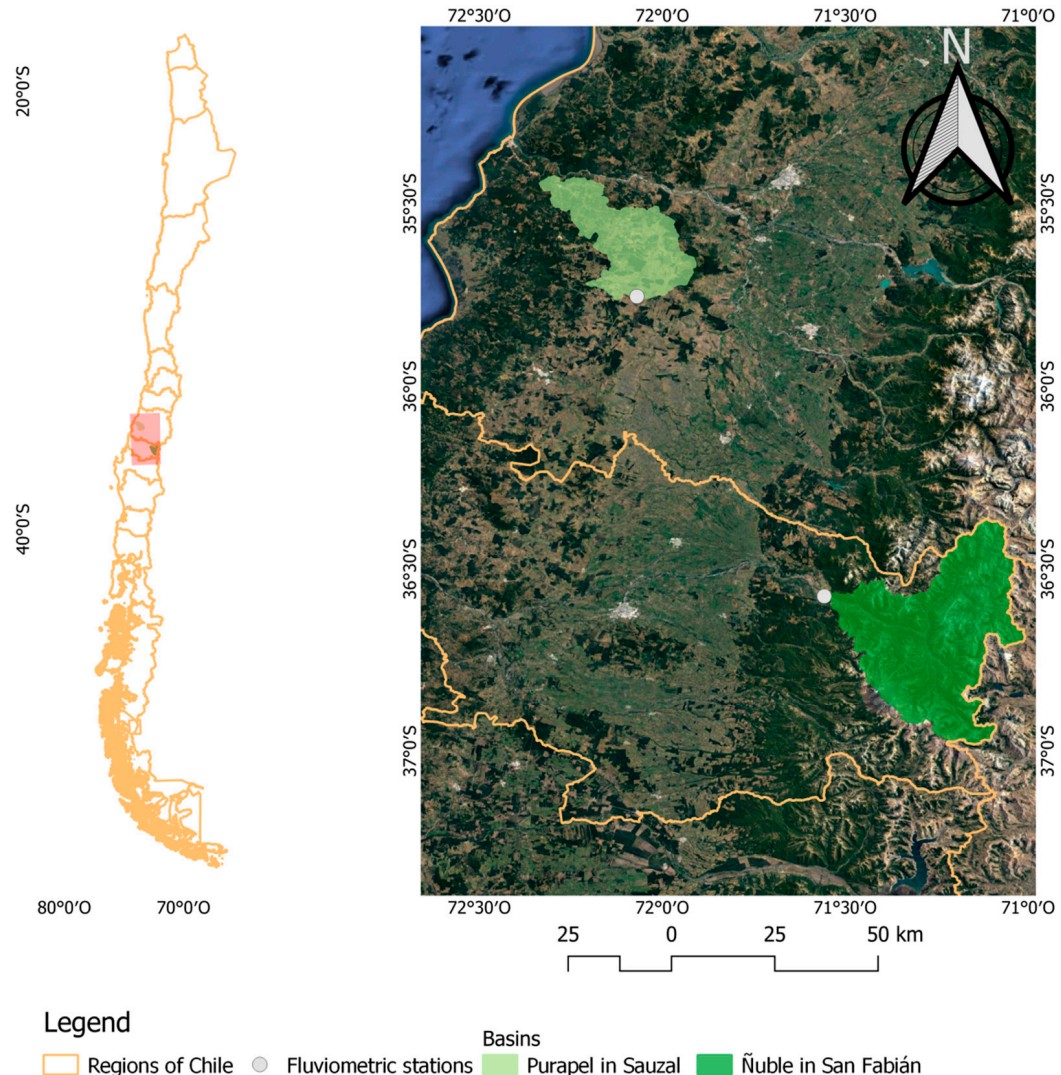

**Figure 1.** Location of the two basins under study within central Chile.

Daily climatic, streamflow, and sediment yield (C) information (1986–2016) was obtained from all gauging stations belonging to DGA at each basin. Additionally, land use (plant cover) changes over time were obtained from the National Forestry Corporation (CONAF), considering years 1986, 1997, 2009, and 2016 (Purapel) and years 1986, 1997, 2008, and 2015 (Ñuble). This was complemented with satellite images (Landsat) and Landsat 5, downloaded from the EarthExplorer server (http://earthexplorer.usgs.gov, accessed on 20 October 2022), for both basins. Furthermore, a topographic, atmospheric, and radiometric correction of the images was performed, following the methodology by [25] and [26]. Subsequently, higher land uses in the areas of interest were identified for each basin under the classifications "no vegetation", "native forest", "plantation", "newly harvested", and "snow", using the supervised classification method available in the open-source software QGIS [27].

*Statistical Analysis*

The fluviometric and sedimentometric (C) data analysis performed in both basins consisted of a temporal behavior's exploratory evaluation of mean daily discharge and temporary C between the years under study. In addition, the relationship between mean discharge and C for the same streamflow was visualized, thus creating sediment transport

curves; see [13] for details. Likewise, differentiated series were created considering summer and winter streamflow to visualize statistical differences between the two seasons. Another factor analyzed was the quotient between winter and summer streamflow, which indicates whether behavior between winter and summer production has remained stable over time.

Sediments' specific concentration for each basin and season was another aspect considered. Thus, the mean annual sediment concentration (mg/L) was transformed into an expression per area unit (mg/L-km$^2$). It was possible to relativize the difference between the basins under study due to this index, which can compare the production per km$^2$ independent of the basin size.

Finally, temporal C trends were assessed using the Mann-Kendall test, which does not require data from a normal distribution and, therefore, can be applied to hydrological information [28,29]. Variables considered were daily concentration, mean annual concentration, mean daily discharge, mean annual streamflow, and the proportion between winter and summer mean streamflow.

For calculation purposes, the Mann-Kendall test and then the Kendall S statistic and its VAR(S) variance are required in the above procedure. Thus, a Z-score is obtained when the sample size is equal to or higher than 8 [30], and its sign and value will determine the trend's orientation and significance, respectively. For the Mann-Kendall S statistic, Equation (1) was used, where the $(x_j - x_k)$ sign function is described in Equation (2), with $x_j$ and $x_k$ being consecutive values of the variable under study. Then, *VAR(S)* is described in Equation (3). Finally, with both values, the Z value was calculated with either formula expressed in Equation (4), depending on the *S* result. The test's critical value for an alpha of 0.05 was defined based on the Z-score, with a positive and negative limit of $-1.96 > Z > 1.96$. In other words, Z values below $-1.96$ and above 1.96 are considered significant for a *p*-value of 0.05.

$$S = \sum_{k=1}^{n-1} \sum_{j=k+1}^{n} sign(x_j - x_k) \tag{1}$$

$$sign(x_j - x_k) \begin{cases} 1 \ if \ x_j - x_k > 0 \\ 0 \ if \ x_j - x_k = 0 \\ -1 \ if \ x_j - x_k < 0 \end{cases} \tag{2}$$

$$VAR(S) \frac{1}{18} \left[ n(n-1)(2n+5) - \sum_{p=1}^{q} t_p (t_p - 1)(2t_p + 5) \right] \tag{3}$$

$$Z = \begin{cases} \frac{S-1}{\sqrt{VAR(S)}} \ if \ s > 0 \\ 0 \ if \ s = 0 \\ \frac{S+1}{\sqrt{VAR(S)}} \ if \ s < 0 \end{cases} \tag{4}$$

## 3. Results

Table 1 shows the mean elevation and slope for both basins. Land use cover proportions are illustrated in Figure 2 and detailed in Table 2 It is observed that vegetation present in the Ñuble basin has remained relatively stable since 1997, doubling the area occupied by native forests and maintaining forest plantations. Moreover, the Purapel basin shows an increased variation, mainly due to the increment in the pine plantations (*Pinus radiata*) in the area. The descriptive analysis of sediment production in the summer (October–March) and winter (April–September) periods for both basins is detailed in Table 2, showing that sediment production is higher in Purapel compared to Ñuble.

**Table 1.** Summary of some characteristics of the basins in this study.

| Basin | Mean Slope (%) | Mean Elevation (masl) |
|---|---|---|
| Purapel | 15.6 | 763.4 |
| Ñuble | 48.8 | 1596.2 |

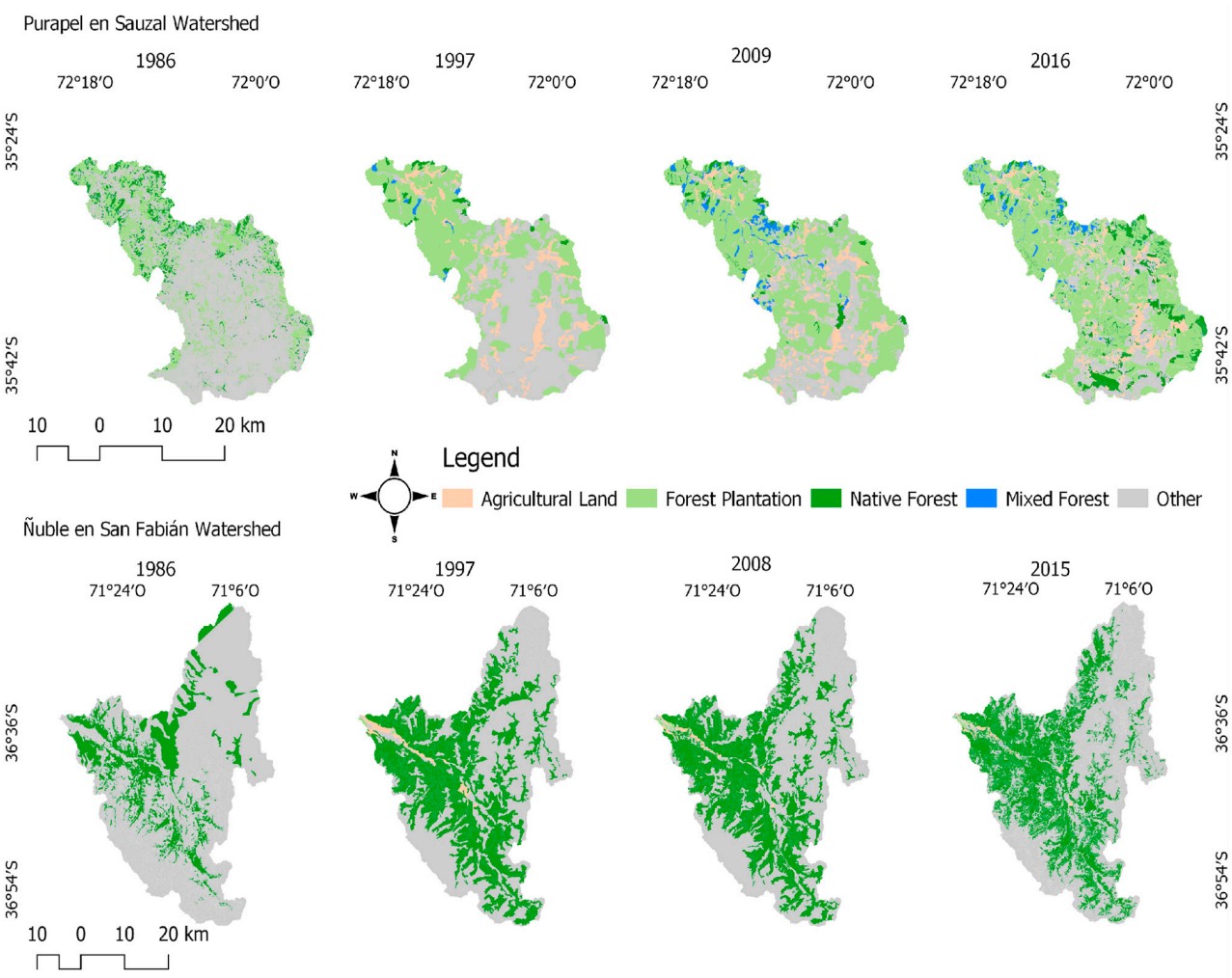

**Figure 2.** Temporal evolution of plant cover in the basins under study.

**Table 2.** Temporal evolution of land uses (%) in the basins under study.

| Type of Cover | Purapel Basin | | | | Ñuble Basin | | | |
|---|---|---|---|---|---|---|---|---|
| | 1986 | 1997 | 2009 | 2016 | 1986 | 1997 | 2008 | 2015 |
| Native Forest | 7.9 | 1.6 | 2.6 | 10.3 | 22.4 | 44.2 | 42.9 | 41.8 |
| Plantations | 23.4 | 42.1 | 55.1 | 58.5 | 0 | 0.1 | 0.3 | 0.2 |
| Agricultural Use | 0 | 9.2 | 8.7 | 7.7 | 0 | 1.3 | 0.7 | 0.6 |
| Mixed forest | 0 | 0.6 | 3.5 | 2.7 | 0 | 0 | 0 | 0 |
| Other Uses | 68.7 | 46.5 | 30.1 | 20.8 | 77.5 | 54.4 | 56.1 | 57.3 |

At the geological level, the parent material (Table 3) of the basins is primarily plutonic in the Purapel basin. These rocks are composed of monzodiorites, granodiorite, monzogranite, and monzonites, among others (MP). In the Ñuble basin, the Volcano–sedimentary parent material predominates, composed of basaltic to dacitic lavas, epiclastic and pyroclastic rocks [31]. Table 4 presents descriptive information for the variables sediment production and flow of each basin. Being Mediterranean basins, the values of these variables present their maximum values during winter.

When evaluating the specific sediment production of both basins (mg/L-km$^2$), greater production of sediments can be observed in the Purapel basin compared to that from the Ñuble basin. In addition, sediment production stabilizes from the year 2000 in the Purapel basin (Figure 3), while the Ñuble basin appears to be relatively stable in the entire studied period (Figure 4).

**Table 3.** Parent material of the watersheds. Own work with data from Sernageomin [31].

| Purapel | | Ñuble | |
|---|---|---|---|
| **Parent Material** | **Area (%)** | **Parent Material** | **Area (%)** |
| Plutonic | 52.9 | Plutonic | 17.9 |
| Volcanic | 17.7 | Volcanic | 21 |
| Metamorphic | 29.1 | Sedimentary continental | 4.7 |
| Sedimentary marine and transitional | 0.3 | Sedimentary volcanic | 56.4 |

**Table 4.** Descriptive data of the variables.

| Basin | Period | Average Sediment Yield (mg L$^{-1}$) | Average Streamflow (m$^3$ s$^{-1}$) | Average Sediment Yield (mg L$^{-1}$ km$^{-2}$) |
|---|---|---|---|---|
| Purapel | Summer | 10.992 | 1.801 | 0.017 |
| | Winter | 29.752 | 5.876 | 0.045 |
| | Annual | 21.534 | 4.163 | 0.032 |
| Ñuble | Summer | 8.337 | 77.065 | 0.005 |
| | Winter | 29.823 | 110.074 | 0.018 |
| | Annual | 19.928 | 93.842 | 0.012 |

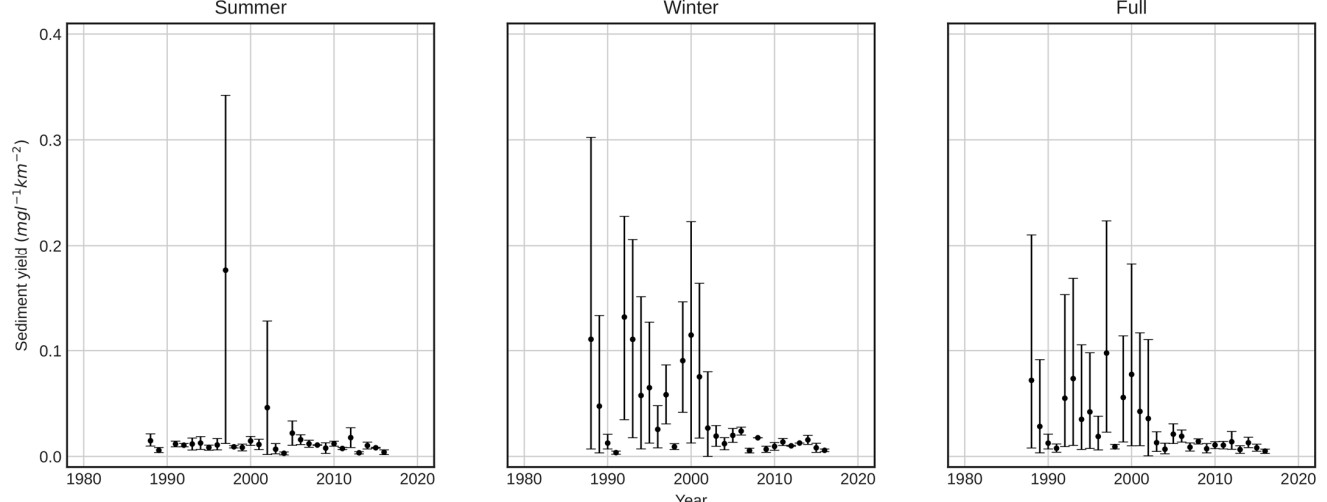

**Figure 3.** Temporal evolution of sediment yield per km$^2$ at Purapel basin. Confidence intervals performed through bootstrapping (*n* = 10,000).

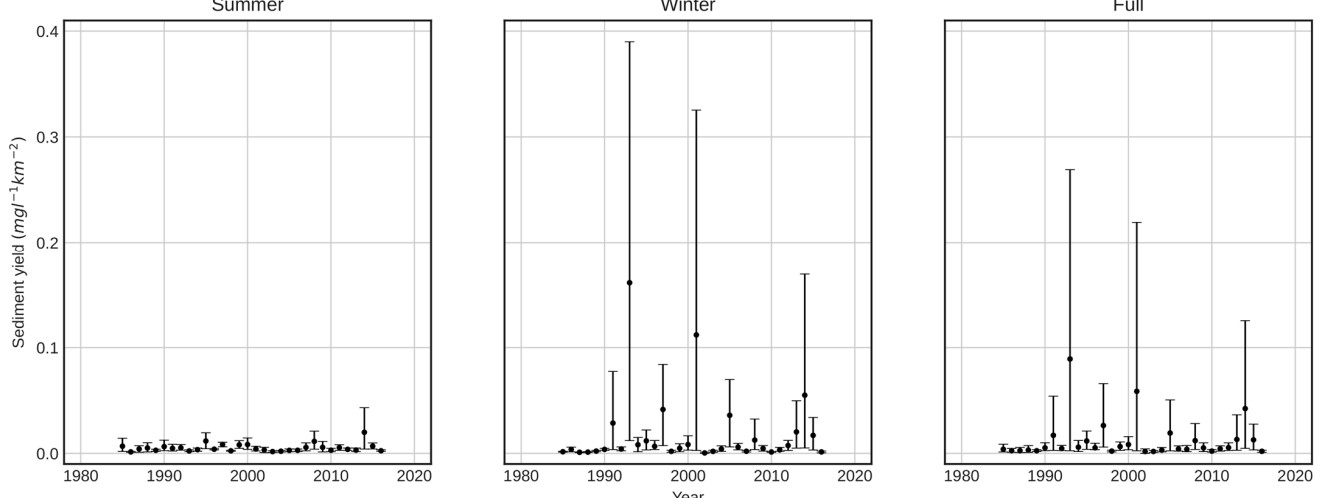

**Figure 4.** Temporal evolution of sediment yield per km$^2$ at Ñuble basin. Confidence intervals performed through bootstrapping (*n* = 10,000).

The relationship between daily sediment concentration and streamflow is illustrated in Figure 5. In the case of Purapel, large value concentration is shown during the summer season, grouped in the 0–4 m³/s and 0–20 mg/L ranges, respectively (with some rare values that significantly escape that range; the reason for those values is unknown). During winter, however, there is a higher variability of concentration, depending on streamflow characteristics. It was not possible to observe a clear relationship in summer and winter between the variables of streamflow and sediment concentration.

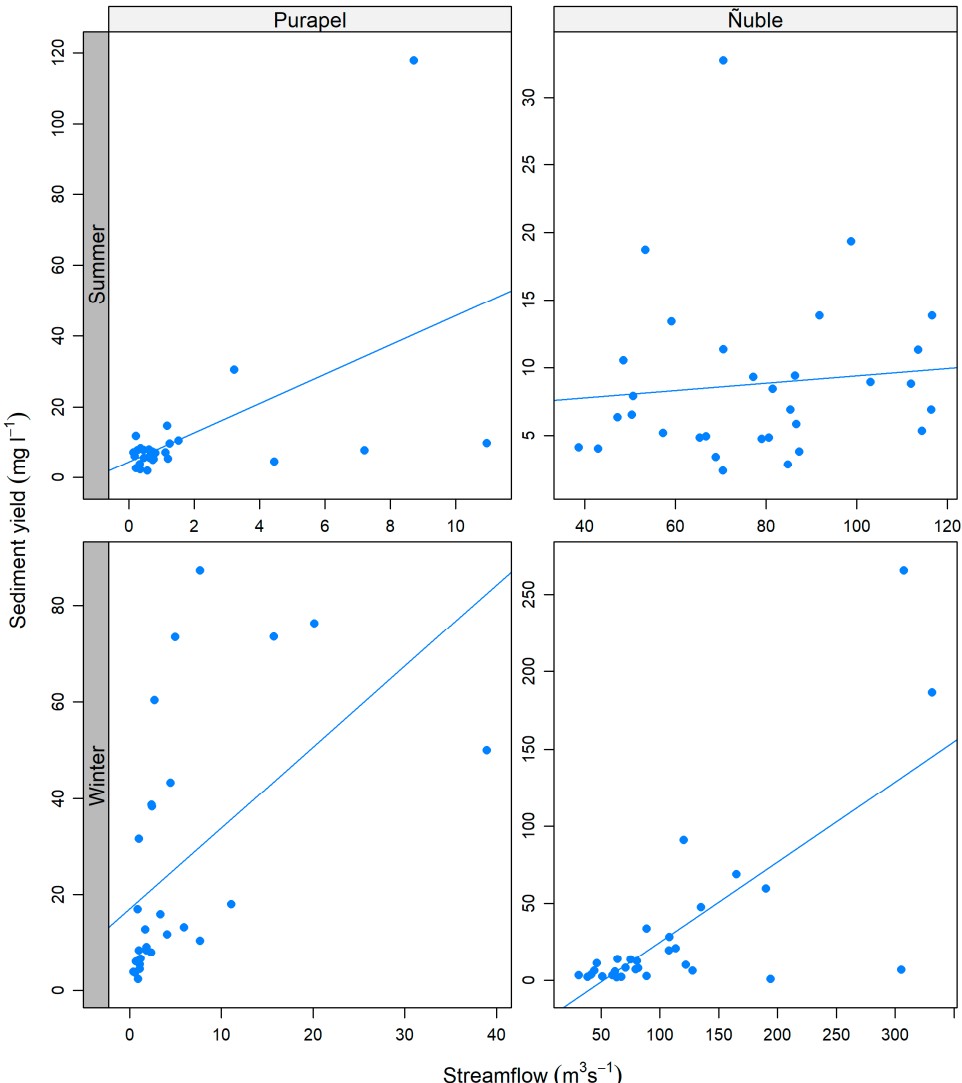

**Figure 5.** Seasonal mean sediment yield and mean streamflow for the Purapel and Ñuble basins.

In the case of Ñuble, a positive relationship between flow and sediment production in winter is verified, although, in a similar way as in Purapel, it manifested value concentration. These are concentrated in the 0–150 m³/s and 0–50 mg/L ranges, respectively, with a growing trend during winter periods. When analyzing both basins, there is generally high sediment variability as a function of streamflow.

Finally, results for the Mann-Kendall Z-values are shown in Table 5. The discriminant value for variable $Z$ ($p < 0.05$) was ±1.96. Therefore, summer sediment production and mean annual streamflow do not show significant temporal trends in either basin. However, the Purapel basin showed negative and significant trends in the study period for winter mean concentration and quotient of mean winter annual sediment yield divided by summer mean annual sediment production.

**Table 5.** Mann-Kendal test results (Z values).

| Period | Variable | Purapel (*Z*) | Ñuble (*Z*) |
|---|---|---|---|
| Summer | Mean Sediment yield | −1.3 | 0.2 |
|  | Mean Streamflow | −0.7 | −1.7 |
| Winter | Mean Sediment yield | −3.3 | 1.2 |
|  | Mean Streamflow | −1.1 | −0.2 |
| Full | Mean Sediment yield | −3.3 | 0.8 |
|  | Mean Streamflow | −1.8 | −1.0 |
| Winter/Summer | Mean Sediment yield | −2.4 | 1.0 |
|  | Mean Streamflow | −0.7 | 0.5 |

## 4. Discussion

Sediment concentration in both basins was significantly higher during winter months. In the case of Ñuble, this value was almost three times larger than that found in summer months, while in the case of Purapel, this value was around 4.5 times higher. Those differences can be explained because the Purapel basin has a total pluvial regime, and streamflow is reduced when the rainy season ceases and can even reach zero l/s during summer, which leads to a reduction in sediment yield during these months. The Ñuble basin has a mixed regimen (rainfall during winter and snowmelt during summer) and large areas without vegetation, meaning that snowmelt during summer months can carry more sediment. In addition, the Ñuble basin cover since 1997 has remained relatively stable, with the native forest being the main cover of this basin, which could explain why sediment yield has remained stable.

From a wider perspective, results show significantly more sediment production in the Purapel basin compared to Ñuble. According to the precipitation totals from both study areas, these results seem contradictory since, on average, the Purapel basin receives only 790 mm/year of rainfall in average. By contrast, Ñuble's precipitation reaches 1409 mm annually (almost double, including snow melting processes). Likewise, Purapel has gentler slopes than those at Ñuble. In this case, plant cover and the soil type present in the basin likely have had a bigger influence on sediment detachment. Thus, native forest and exotic plantations in the Purapel basin reach almost 70% of the total area, while that proportion is 43% in Ñuble.

Moreover, when analyzing winter sediment yields over time in the Purapel basin, it seems to show a negative trend, as sediment production decreased between 1990 and 2005 and remained relatively stable until 2016. The winter season does not seem to show any trend in the case of the Ñuble basin. In addition, when evaluating the proportion between winter and summer sediment yield, it is observed that this quotient tends to decrease in the Purapel basin, which could explain a fall in the release of winter sediment production.

This quotient (winter/summer) indicates a relatively stable relationship over time in the Ñuble basin, except for two isolated events, showing no significant temporary changes in the ratio of sediment production between both seasons. Additionally, none of the variables analyzed showed a marked negative and significant trend, an indication that sediment emissions have remained stable over time (there is actually some incremental but not significant values). This evidence could be related to land uses, as since 1997, they have remained relatively stable over time (Table 2), especially for native forest uses.

Likewise, it should be noted that there are differences in the slope and average height of the studied basins (Table 1), being greater in the Ñuble basin and, therefore, greater production of sediment in this basin is to be expected. However, the specific sediment production (sediment per km$^2$) of the Purapel basin is greater than that of Ñuble (Table 4), and this could be explained by the changes in coverage that the Purapel basin has undergone.

The intensity of precipitation is a relevant factor when evaluating the production of sediments in a basin, derived from the fact that at higher intensities of precipitation, the erosivity of rain is greater [32]. Sangüesa et al. [33] analyzed the trend of rainfall intensity

in the Maule Region, finding that it has remained relatively stable for all its durations. In the case of Ñuble, since there was no similar study, the trends of two stations with at least 30 years of information were analyzed, evaluating their maximum intensities in 24 h. The results of this test show that both seasons show significant changes (San Fabián $Z = -1.4$, Caracol $Z = -1.7$).

These results indicated that the main temporal change in the basins under study corresponds to land use and, therefore, this factor seems to be the most relevant in explaining the production of sediments in the studied basins.

Moreover, Pepin et al. [34] studied the relationship between plant cover, slope, and climate in sediment production from 66 Andean basins in Chile, concluding that there is no significant correlation between plant cover and sediment production. However, Pizarro et al. [16] verified the stabilizing effect of vegetation on the basin riverbed in a study conducted at the Purapel river basin (Nirivilo station), finding a reduction in the width and number of annual sediment–discharge curves documented at that location. This difference in the results from both studies may be due to the methodologies used in the research since Pepin et al. [34] found no statistically significant relationships in the group of analyzed basins, though this does not imply that vegetation is not stabilizing them, but rather that they are not the main factor in the stability of the basins. In this context, the authors recommend replicating the methodology applied by Pizarro et al. [16] for the basins analyzed by Pepin et al. [34] to quantify the effect of vegetation on these basins (but at an individual level). Moreover, the trend test results showed a decrease in sediment emissions in the Purapel basin during winter periods, which may be due to the stabilizing role generated by forest plantations over time that have increased their presence in the basin from 20% (1986) to more than 50% (2016) [16]. This is endorsed by authors focusing their research on basins under Mediterranean climates, such as Rius et al. [35], who analyzed a *Pinus Sylvestris*-covered basin in Spain, concluding that this cover provides stability to the drainage área. Similarly, Abdelwahab et al. [36] simulated sediment transport under different scenarios, finding that the change in land use from agriculture to forest reduced sediment production. In addition, Spalevic et al. [37] found an inverse relationship between forested areas and sediment yield inside the studied basins. Spalevic et al. [37] verified the effect of plant cover in the Miocki Potok basin (Montenegro), finding that a change (from agricultural to forest land) of eight percentage points decreases sediment production by 14% and peak flows by 3.5%. In addition, Duran-Llacer et al. [38] verified that vegetation has a stabilizing role within the Lanjarón basin (Spain), reducing the variability of flows and sediment production within the basin. However, the authors also pointed out that the main factor in the variability and production of sediments is the intensity of rainfall. Another factor to consider is the variable nature of Mediterranean climates, derived from the fact that there are subtypes of Mediterranean climates influenced by atmospheric variables. In this context, Peña-Angulo et al. [39] identified the influence of weather patterns on streamflow, erosion, and sediment yield in Mediterranean basins located in France, Greece, Italy, Israel, Morocco, Portugal, Spain, Slovenia, and Tunisia, finding differences depending on the climate type and geography of the area.

Among the limitations of the study, there is the lack of daily sedimentation data and sub-hourly rainfall information in the basins; therefore, it is not possible to quantify the effect of maximum precipitation intensities on sediment production, with a loss of relevant spatial information to analyze the behavior of sediment production within the study sites.

On the other hand, the Ñuble basin, being considerably larger and having a different slope, average altitude, parent material, and fluvial regime (compared to the Purapel basin), makes it difficult to develop a comparative analysis of both basins. Under ideal conditions, it is advisable to compare basins with similar characteristics to control the effect of the variables. However, it is important to note that most of these factors are stable over time, with the exception of land uses that can vary considerably, and our results show that their influence is relevant when estimating sediment production in the basins.

## 5. Conclusions and Recommendations

Based on the obtained results, it is possible to conclude that the production of sediments in the studied areas was greater during winter months (even in the basin with a rainfall regime) because the basins under study are located under a Mediterranean climate. However, there was a significant reduction in sediment yield in the Purapel basin, and that could be an effect of the stabilization of the watershed after the significant increase of pine forest plantations over time, while the Ñuble basin did not show a significant reduction in sediment production, despite having increased the area of native forest, a fact that could be due to the fact that this is vegetation without forest management.

This study provides important findings for future land management decisions in the Mediterranean climates of central Chile. For the first time, the behavior of two watersheds is contrasted (one covered with native forest and the other one with forest plantations), appreciating that the latter presents a reduction in sediment production over time, which means that forest plantations are efficient in sediment retention, even in contrast to native forest. However, both basins have different types of soil, topography, etc., meaning that more studies are needed to support this theory.

In the last period, Chile has suffered wildfires that have devoured thousands of hectares of forests, leaving soil susceptible to the effects of erosion and sedimentation, which affects hydrological balances. Under the above context and based on the results achieved, the authors of this study recommend reforesting the areas that have lost their tree cover due to wildfires, as well as those devoid of vegetation for other reasons (indiscriminate clearcutting, habilitation of agricultural land, among others). This is derived from the fact that vegetation protects the soil, mitigating the erosive effects of precipitation, thus reducing sediments being transported by rivers and stabilizing the basin.

It is worth mentioning that for soil protection purposes, native or exotic species can be used and that this decision is related to other aspects. One of them is the fragility of the ecosystem and the quality of the soil in its ability to support vegetation; if the soil is highly degraded, it will be recommended to use species that are resistant to poor soils, as is the case with the exotic species that are used in Chile. In this case, it is recommended to use fast-growing species that promote early protection and stabilization of the soil and banks within the basin. Thus, the exotic species used in the country have shown a stabilizing effect in the basins, as long as water resources are not affected at a basin level. Chile is a country that is quickly drying out, so water must be a factor to consider in any land management decision, with no exception. Moreover, forested areas in Mediterranean climates can indeed increase or decrease water availability, depending on where they are, so rigorous hydrological models must be developed before moving forward with afforestation initiatives.

Another aspect to consider is the ecosystem services of native forests. In this sense, planting native species allows for maintaining or improving biodiversity (flora and fauna) and, with it, improves the landscape and increases resistance to wildfires, among others. For this reason, it is advisable to afforest with native species in the areas where they predominated before the fires, avoiding the replacement of native forests by exotic species and promoting the recovery of native vegetation in degraded basins.

**Author Contributions:** Conceptualization: R.P., A.I. and P.G.-C.; methodology: R.P., A.I., C.S. and P.G.-C.; software: B.I. and A.I.; validation: R.P.; formal analysis: C.S., A.I., J.P, R.M. and C.V.; investigation: R.P., P.G.-C., A.I., C.S. and M.V.; resources: R.P.; data curation: B.I., A.I., J.P.F., R.B.-O. and F.P.; writing—original draft preparation: R.P., C.S., P.G.-C., F.B. and A.I.; writing—review and editing: P.G.-C., R.P., G.M. and A.I.; visualization: A.I., J.P. and B.I.; supervision: R.P.; project administration: R.P. All authors have read and agreed to the published version of the manuscript.

**Funding:** This research was funded by ANID BASAL FB210015.

**Institutional Review Board Statement:** Not applicable.

**Informed Consent Statement:** Not applicable.

**Data Availability Statement:** Most data used in this study can be found at DGA's, USGS and CONAF websites (www.dga.cl, http://earthexplorer.usgs.gov, and www.conaf.cl, all accessed on 8 November 2022).

**Acknowledgments:** The authors deeply thank the ANID BASAL Center (FB210015), to the doctoral scholarship ANID-PFCHA/2021-21210861, and the Center for Mining Sustainability, a joint adventure between Colorado School of Mines and Universidad Nacional de San Agustín de Arequipa.

**Conflicts of Interest:** The authors declare no conflict of interest.

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
