# Peer review of "Establishment of Monterrey Pine (Pinus radiata) Plantations and Their Effects on Seasonal Sediment Yield in Central Chile"

_sustainability, doi:10.3390/su15076052_

Round 1
Reviewer 1 Report
I appreciate the chance to review the manuscript "Establishment of Monterrey pine (Pinus radiata) plantations and their effects on seasonal sediment yield in central Chile by Pizarro et al. It was an interesting look at the effect of land use on sediment yield compared to other variables within these two sedimentary basins.
This was an interesting concept/hypothesis, but I'll admit that it left me a bit confused and underwhelmed - although I think with some reworking it will hopefully make a compelling study. I wish the authors success in this endeavor! Please let me know if you have any additional questions or need any further clarification, I am more than happy to help.
My main points of concern are as follows:
The hypothesis states that sediment yield is a multifactorial process, but there is very little mention of the of the other factors - the authors compare the two basins, but offer little insight into the characteristics of each, how comparable are these two areas specifically? There is little to mention of elevation, slope/ gradient, soil, and parent material - all of which play a large role in sediment load.
Fluvial style also matters greatly when it comes to sediment yield – both from an erosional and depositional point of view, I don’t think there was any mention of this – are these fluvial systems comparable when thinking about sediment moving within the basin?
Additionally, I think some note and/or consideration needs to be made to address that the Ñuble basin is approximately 2.5x the size of the Purapel basin, but the total sediment yield between them is similar – that’s interesting, and I think part of the story of how sediment is being moved in these basins, but wouldn’t this also affect the sediment yield per km2 values?
One of my biggest concerns is the land use around the rivers and tributaries themselves, looking at land use within the entire basin is worthwhile, but if the Purapel is surrounded by farm fields and the Ñuble is surrounded by forests, then it would be intuitive that the sediment yield would be higher in the Purapel. I would think unchanging forest land cover would seemingly play little role in sediment yield if these forests were distal from the channel moving the sediment. I think some clarification, a sentence or paragraph more on the land use (and land use change) proximal to the fluvial system would be worthwhile.
The climate of the Purapel is noted to be Mediterranean but no mention of the climate of the Ñuble basin outside of the annual precipitation, is it also Mediterranean?
Some areas that I think need to be reworded or clarified - In the results, and at various other places, there is focus on the fact that there is no change in sedimentation in the Ñuble basin as vegetation has remained relatively stable since 1997, with the 10 years prior to that seeing a doubling of the vegetation, which is significant. Although I think some note should be made that the sediment yield across this time is already relatively low, even given that streamflow is so much higher, almost 20x (!) than at Purapel. Again, this gets at the heart of the matter for me that I have no idea if these basins are even comparable – so a note on the fluvial system and gradient/topography and perhaps a statement on why they are comparable is needed.
Similarity, at line 200 it is stated that a greater production of sediments is observed in the Purapel basin – is this the total sediment yield or the sediment yield per km2? And lines 203-204 state that there is “variability during the summer that remains over time and stays relatively constant” – what is this saying exactly if it is both variable and constant? Is this to say annually from summer to summer?
There is mention of some “rare values that significantly escape that range” in lines 218-219, I think this bears at least a mention of why this occurs, what looks to be 7 events since 1997 – there appears to be a something occurring during these events that can be expanded upon. What is causing these pulses of sediment – climate/weather, field clearing, logging? If the reason here is something that affects both basins, then you can reinforce the idea that the establishment of plantation forests resulting in reduced sediment yields.
Perhaps some of my confusion here is just the need for a more clarified paragraph just pointing to the abrupt drop in sediment yield post-2002 at Purapel that is due to the establishment of the Pinus plantations – I think this is the key point being made here at least, but that connection is not clear or apparent from my first reading of this manuscript, so some clarification is needed if this is indeed the case.
Line 227 - The Ñuble basin summer relationship between flow and sediment production does not appear very clear in figure 5, some note or clarification here would be helpful.
Lines 251-252 it is noted that forest types haven’t changed over time at Ñuble, but from 1987-1997 they almost doubled, so need to be clearer here on the timing of this stability.
Line 251 – sediment in place of sediments
Line 258-259 – the word “denotes” here does not make sense in this context.
Line 260 – “Thus” does not makes sense in this context.
Line 263-264 – The winter decrease at Purapel seems to occur very abruptly starting in 2002 – is there a potential reason for this?
Line 266 – Is the recent increase truly a change in sediment yield or just noise? 2013-2015 appear higher, but 2016 returns to the average low values again.
275 – I am not sure this is true about no unchanged land use over time? Clarify if you mean since 1997 as native forest has remained relatively constant, although agriculture, while low, has been halved.
Line 294 - Are the plantations replacing native forests? Agriculture land? “Other uses”? I think some clarification of what land is being replaced here by these native forests? Also, a sentence, perhaps earlier on, explaining what constitutes the “other uses” category of land cover.
Line 315 – I’m not sure the timing here matches with the increase in Pinus radiata occurring before the drop in sediment yield in 2003 – is this a function of the forest establishing itself? If so, you should clarify that here.
Author Response
I appreciate the chance to review the manuscript "Establishment of Monterrey pine (Pinus radiata) plantations and their effects on seasonal sediment yield in central Chile by Pizarro et al. It was an interesting look at the effect of land use on sediment yield compared to other variables within these two sedimentary basins.
This was an interesting concept/hypothesis, but I'll admit that it left me a bit confused and underwhelmed - although I think with some reworking it will hopefully make a compelling study. I wish the authors success in this endeavor! Please let me know if you have any additional questions or need any further clarification, I am more than happy to help.
Many thanks for your interest in this study and willingness to provide positive and encouraging feedback. This is greatly appreciated
My main points of concern are as follows:
The hypothesis states that sediment yield is a multifactorial process, but there is very little mention of the of the other factors - the authors compare the two basins, but offer little insight into the characteristics of each, how comparable are these two areas specifically? There is little to mention of elevation, slope/ gradient, soil, and parent material - all of which play a large role in sediment load.
We have included a number of statements throughout the document now to emphasize that there are other factors involved and what these factors are, however we also emphasize that over time, many of these factors are static, therefore not necessarily related to some of the variable behavior that we outline in the paper. We have also included new tables to detail some of these characteristics such as parent material and topography so the reader can better understand the study area. We are also clear that there are limitations to this study and that further research is needed, but that this work provides an important first step towards determining the underlying factors.
Fluvial style also matters greatly when it comes to sediment yield – both from an erosional and depositional point of view, I don’t think there was any mention of this – are these fluvial systems comparable when thinking about sediment moving within the basin?
We have added some more details regarding the fluvial regimens and detailed the similarities and differences between the two basins considered in this study
Additionally, I think some note and/or consideration needs to be made to address that the Ñuble basin is approximately 2.5x the size of the Purapel basin, but the total sediment yield between them is similar – that’s interesting, and I think part of the story of how sediment is being moved in these basins, but wouldn’t this also affect the sediment yield per km2 values?
We have added some text to address this, and included discussion of the other factors and recognize that although this makes a comparative analysis more challenging, many of these other factors are stable over time, and hence helps with understanding the effects of these variables.
One of my biggest concerns is the land use around the rivers and tributaries themselves, looking at land use within the entire basin is worthwhile, but if the Purapel is surrounded by farm fields and the Ñuble is surrounded by forests, then it would be intuitive that the sediment yield would be higher in the Purapel. I would think unchanging forest land cover would seemingly play little role in sediment yield if these forests were distal from the channel moving the sediment. I think some clarification, a sentence or paragraph more on the land use (and land use change) proximal to the fluvial system would be worthwhile.
We have have added some text about land use changes and how it can vary considerably and that our results show their relevance to estimating sediment production in the basins.
The climate of the Purapel is noted to be Mediterranean but no mention of the climate of the Ñuble basin outside of the annual precipitation, is it also Mediterranean?
We have addressed this by adding details of Ñuble’s climate.
Some areas that I think need to be reworded or clarified - In the results, and at various other places, there is focus on the fact that there is no change in sedimentation in the Ñuble basin as vegetation has remained relatively stable since 1997, with the 10 years prior to that seeing a doubling of the vegetation, which is significant. Although I think some note should be made that the sediment yield across this time is already relatively low, even given that streamflow is so much higher, almost 20x (!) than at Purapel. Again, this gets at the heart of the matter for me that I have no idea if these basins are even comparable – so a note on the fluvial system and gradient/topography and perhaps a statement on why they are comparable is needed.
We have rewritten a number of paragraphs through the document to try and make them clearer. As you mention, there is a running theme through the feedback and something that we have tried to address in previous comments and something that we recognize could be considered as a potential limitation. We have added statements to make it clear what our assumptions are with regards to the other variables involved, and how these would necessarily impact the comparative analysis. The significant aspect being the stability of these variables over time.
Similarity, at line 200 it is stated that a greater production of sediments is observed in the Purapel basin – is this the total sediment yield or the sediment yield per km2? And lines 203-204 state that there is “variability during the summer that remains over time and stays relatively constant” – what is this saying exactly if it is both variable and constant? Is this to say annually from summer to summer?
We have addressed both of these parts to make it explicit as to what is meant
There is mention of some “rare values that significantly escape that range” in lines 218-219, I think this bears at least a mention of why this occurs, what looks to be 7 events since 1997 – there appears to be a something occurring during these events that can be expanded upon. What is causing these pulses of sediment – climate/weather, field clearing, logging? If the reason here is something that affects both basins, then you can reinforce the idea that the establishment of plantation forests resulting in reduced sediment yields.
We have added some text to this section to emphasize that we recognise there are some values out of range and that the reasons for this have been studied, but we are still unsure as to the cause or origin of this.
Perhaps some of my confusion here is just the need for a more clarified paragraph just pointing to the abrupt drop in sediment yield post-2002 at Purapel that is due to the establishment of the Pinus plantations – I think this is the key point being made here at least, but that connection is not clear or apparent from my first reading of this manuscript, so some clarification is needed if this is indeed the case.
We have edited the text so that this key point is much more explicit.
Line 227 - The Ñuble basin summer relationship between flow and sediment production does not appear very clear in figure 5, some note or clarification here would be helpful.
We have added a more thorough description of the behaviors of the relationships and updated the graphs to make it clearer as to how we interpreted these graphs.
Lines 251-252 it is noted that forest types haven’t changed over time at Ñuble, but from 1987-1997 they almost doubled, so need to be clearer here on the timing of this stability. Addressed.
Line 251 – sediment in place of sediments. Addressed.
Line 258-259 – the word “denotes” here does not make sense in this context. Addressed.
Line 260 – “Thus” does not makes sense in this context. Addressed.
Line 263-264 – The winter decrease at Purapel seems to occur very abruptly starting in 2002 – is there a potential reason for this?
We have added some text to address the abrupt decrease.
Line 266 – Is the recent increase truly a change in sediment yield or just noise? 2013-2015 appear higher, but 2016 returns to the average low values again.
We have removed the reference to an increase.
275 – I am not sure this is true about no unchanged land use over time? Clarify if you mean since 1997 as native forest has remained relatively constant, although agriculture, while low, has been halved.
We have updated the text to be more explicit by referring directly to native forests.
Line 294 - Are the plantations replacing native forests? Agriculture land? “Other uses”? I think some clarification of what land is being replaced here by these native forests? Also, a sentence, perhaps earlier on, explaining what constitutes the “other uses” category of land cover. Addressed.
Line 315 – I’m not sure the timing here matches with the increase in Pinus radiata occurring before the drop in sediment yield in 2003 – is this a function of the forest establishing itself? If so, you should clarify that here. Addressed.
Reviewer 2 Report
Comment to the author
‘Establishment of Monterrey pine (Pinus radiata) plantations and their effects on seasonal sediment yield in central Chile’
General
I understand the sophistication of this study and I believe a lot of energy had been devoted to the work.
The Abstract is nicely composed. The Introduction provides state-of-the-art references and bases for the study whereas readers can easily understand that sediment yield is the result of a multifactorial process in which plant cover and hydrological regimens could be a determining factor. The Materials and Methods part is clear enough in detail and elaboration, although I recommend the author to add details or further elaborate on the study area in the form of a table. The Results and Discussion are nicely composed as well. The Conclusion is in accordance with the research goals. Regarding the Reference, though, I suggest the authors to add relevant reference on different types of plant covers (forest plantations and native forest) sediment yield.
Overall, substantially, this manuscript is nicely presented. However, I suggest the author to revise the manuscript as suggested to better improve it.
Specific
I made some comments and revision recommendations in the manuscript by track change for the consideration of the authors to improve their manuscript.
Comment 1
Line 107. Suggest the authors to add relevant reference on different types of plant covers (forest plantations and native forest) sediment yield.
Comment 2
Line 108. ……plant cover……
and hydrological regimens ?
Comment 3
Recommend the author to add details (slope, altitude, temperature, plant covers, streamflow and sediment yield information)or further elaborate on the study area in the form of a table.
Comment 4
Line 187. Figure 2. Why are the years of the both basins different?
Comment 5
Table 2, Sediment yield, Streamflow, and Sediment yield per km2 of the both basins may be better described by year.
Comment 6
The summer sediment yield per km2 of Purapel basin was particularly high in 1997. Please explain this reason.
Comment 7
What is the novelty of the paper, what is new in this study that were not yet explored / examined / explained by others?
Comment 8
What is the importance of this study?
Comment 9
How do the authors think this study differs from already published studies? What makes this study a critical article?
Comment 10
Highlight the most important results especially the unique ones/novelty and make an in-depth discussion about these important results.
Comment 11
What is the take home message that will imprint to the minds of the readers after reading the paper?
Comment 12
The above comments should be carefully checked and identified inconsistences should be corrected throughout the manuscript.
Author Response
I made some comments and revision recommendations in the manuscript by track change for the consideration of the authors to improve their manuscript.
We appreciate the reviewer taking the time to make changes to the document, unfortunately however, we never received that document, even though we explicitly requested it from the journal. However, together with the changes suggested by three other reviewers, we hope this version is good enough
Comment 1: Line 107. Suggest the authors to add relevant reference on different types of plant covers (forest plantations and native forest) sediment yield.
We have added a number of references to address this
Comment 2: Line 108. ……plant cover…… and hydrological regimens ?
We have added some text including information about hydrological regimens
Comment 3: Recommend the author to add details (slope, altitude, temperature, plant covers, streamflow and sediment yield information)or further elaborate on the study area in the form of a table.
We have added tables and some text to address this comment
Comment 4: Line 187. Figure 2. Why are the years of the both basins different?
These were the years when data was available
Comment 5: Table 2, Sediment yield, Streamflow, and Sediment yield per km2 of the both basins may be better described by year.
We have made this change in the document
Comment 6: The summer sediment yield per km2 of Purapel basin was particularly high in 1997. Please explain this reason. Addressed
Comment 7: What is the novelty of the paper, what is new in this study that were not yet explored / examined / explained by others?
We have addressed this with some additional notes at the end of the Conclusions
Comment 8: What is the importance of this study?
This is encompassed in the changes we made to Comment 7
Comment 9: How do the authors think this study differs from already published studies? What makes this study a critical article?
This study talks specifically about forested areas located under Mediterranean climates of central Chile and we believe the first of its kind to address this in the specific study area
Comment 10: Highlight the most important results especially the unique ones/novelty and make an in-depth discussion about these important results.
We have addressed this with some additional text in the discussion and conclusions
Comment 11: What is the take home message that will imprint to the minds of the readers after reading the paper?
Again, this is addressed as part of the answer to Comments 9 & 10
Comment 12: The above comments should be carefully checked and identified inconsistences should be corrected throughout the manuscript.
We have reworked significant parts of the document to take into account the comments received. We identified various inconsistencies that we corrected in this new version of the document.
Reviewer 3 Report
The article is well-writtwn, in an understandable language. The references are accurate, the results are clearly descirbed. However, I think the visual side can be improved: the texts on the figures are a bit murky, and maybe the data of table 1 (or even table 2) could be presented in a diagram. But overall, its an easy-to-read article on a topic which is very important in today land and soil management.
Author Response
The article is well-writtwn, in an understandable language. The references are accurate, the results are clearly descirbed. However, I think the visual side can be improved: the texts on the figures are a bit murky, and maybe the data of table 1 (or even table 2) could be presented in a diagram. But overall, its an easy-to-read article on a topic which is very important in today land and soil management.
The authors deeply thank the reviewer for taking the time to look at our manuscript. We have taken on board this feedback and updated the figures and tables, and tried to make them more clearly presented.
Reviewer 4 Report
The manuscript titled Establishment of Monterrey pine (Pinus radiata) plantations and their effects on seasonal sediment yield in central Chile is focused on the statistical evaluation of sediment yield by the establishment of plantations of Monterrey pine. However, information on the characteristics of sediments and soil, e.g., granulometry, is missing.
The authors should check for typos.
Author Response
The manuscript titled Establishment of Monterrey pine (Pinus radiata) plantations and their effects on seasonal sediment yield in central Chile is focused on the statistical evaluation of sediment yield by the establishment of plantations of Monterrey pine. However, information on the characteristics of sediments and soil, e.g., granulometry, is missing.
The authors sincerely thank the reviewer for taking the time to see our manuscript. We have addressed this comment by adding additional tables and text to the document to help the reader better understand and appreciate the characteristics of the basin.
The authors should check for typos.
We have undertaken a revision of the document, corrected a number of typos and improved the writing of many parts of the text.
Round 2
Reviewer 1 Report
First let me start by saying thank you to the authors and commending them for the effort and time they have taken to address my comments and concerns - I certainly appreciate the thoughtful responses to each of my many inquiries. After reading through the revisions and responses, I believe the authors have added clarity to this paper, and a nice research study is presented here. I am satisfied with the manuscript in its current form.
If you have any questions or comments, feel free to contact me.
Author Response
Thank you very much!
Reviewer 2 Report
I read and checked the revised version of the manuscript and found that the authors already revised the manuscript accordingly, thus the ms is very much improved now. Therefore I recommend this ms is suitable to be published in this reputable Sustainability journal.
Author Response
Thank you very much!